# Mitigating Error Accumulation in Continuous Navigation via Memory-Augmented Kalman Filtering

**Yin Tang** [1 2]  **Jiawei Ma** [2 *]  **Jinrui Zhang** [3]  **Alex Jinpeng Wang** [3]  **Deyu Zhang** [3 †]

## Abstract

Continuous navigation in complex environments is critical for Unmanned Aerial Vehicle (UAV). However, the existing Vision-Language Navigation (VLN) models follow the dead-reckoning, which iteratively updates its position for the next waypoint prediction, and subsequently construct the complete trajectory. Then, such stepwise manner will inevitably lead to accumulated errors of position over time, resulting in misalignment between internal belief and objective coordinates, which is known as "state drift" and ultimately compromises the full trajectory prediction. Drawing inspiration from classical control theory, we propose to correct for errors by formulating such sequential prediction as a **recursive Bayesian state estimation** problem. In this paper, we design **NeuroKalman**, a novel framework that decouples navigation into two complementary processes: a *Prior Prediction*, based on motion dynamics and a *Likelihood Correction*, from historical observation. We first mathematically associate Kernel Density Estimation of the measurement likelihood with the attention-based retrieval mechanism, which then allows the system to rectify the latent representation using retrieved historical anchors without gradient updates. Comprehensive experiments on TravelUAV benchmark demonstrate that, with only 10% of the training data fine-tuning, our method clearly outperforms strong baselines and regulates drift accumulation. The code will be publicly released at https://github.com/yinntag/Neuro-Kalman.

---

*Project Leader. [1]Big Data Institute, Central South University, Changsha, China. Work done while working at CityUHK as a visiting scholar. [2]Department of Computer Science & Institute of Digital Medicine, City University of Hong Kong, Hong Kong, China [3]School of Computer Science, Central South University, Changsha, China. Correspondence to: Deyu Zhang <zdy876@csu.edu.cn>.

*Proceedings of the 43rd International Conference on Machine Learning*, Seoul, South Korea. PMLR 306, 2026. Copyright 2026 by the author(s).

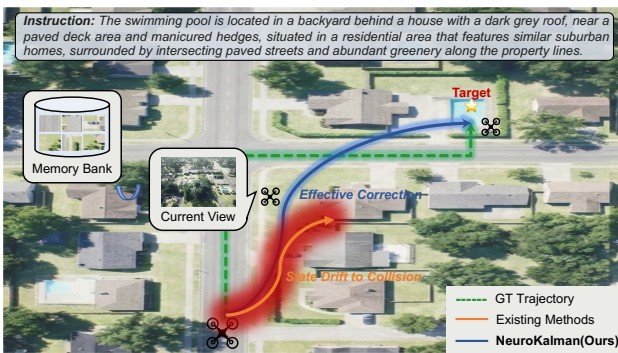

*Figure 1.* **Illustration of state drift mitigation.** Given a global instruction, existing models ignore the history but make prediction only from current inputs, and thus suffer from accumulated error and state drift to collision (*orange line*). Instead, our NeuroKalman framework introduces a Kalman correction mechanism by fusing historical measurements as anchors for prediction to rectify the trajectory prediction (*blue line*).

## 1. Introduction

Continuous navigation is fundamental to achieving full automation in Unmanned Aerial Vehicles (UAVs) (Liu et al., 2023; Lee et al., 2025). Recent advance has been made to utilize Vision-Language Navigation (VLN) models, which follow a global natural language instruction and the local visual observation at the current step, to predict the next waypoint (Fan et al., 2023; Wang et al., 2024b; Gao et al., 2025). By incrementally updating the position from past estimates for the next forecasting, the model predicts the full trajectory for UAV navigation step-by-step, which draws a parallel to the concept of dead-reckoning in control theory.

Nevertheless, such mechanisms are sensitive to individual outlier predictions (Wang et al., 2024b;a; Cai et al., 2025) and will accumulative error over time. As illustrated in Figure 1, without a dedicated correction mechanism, the disparity between the UAV position and the expected trajectory intensifies. As the linguistic instruction remains unchanged throughout the navigation (Liu et al., 2023), it is then assumed to already indicate a global plan implicitly from the initial position to the target destination. Then, the disparity, between the objective UAV coordinates and the expected (internal) positional belief, a concrete instantiation

to the concept of "state drift", will cause a fundamental misalignment in the latent space between current visual observation and the initial language instruction for subsequent waypoints prediction, which further magnify the error in navigation and hurts model generalization (Krantz et al., 2020; Georgakis et al., 2022; Chen et al., 2022).

Draw inspiration from classical Bayesian state estimation (Särkkä & Svensson, 2023), we consider a recursive feedback loop where the historical measurements are used to periodically correct internal predictions. In detail, it consists of two complementary phases: a *Prediction* step that generates a prior estimate based on motion dynamics, and an *Update* step that rectifies this belief using the likelihood of historical measurements. We note that this diverges from existing models (Wang et al., 2024b; Lin et al., 2025) which primarily focus on generating strong prior estimation while overlooking an explicit mechanism to calibrate these predictions (Kloss et al., 2021; Revach et al., 2022).

To this end, we propose NeuroKalman, which directly models the Bayesian prediction-update cycle among the latent representations, and decomposes the navigation into two streams. For the *Prediction*, we employ a Recurrent Neural Network (RNN) to model the motion dynamics for initial forecasting. Then, with the mathematical association between Kernel Density Estimator (KDE) over the measurement likelihood (Katharopoulos et al., 2020) and the attention mechanism (Vaswani et al., 2017), we introduce an Memory Bank to integrate historical observation in the *Update*, which can be jointly used to estimate Kalman Gain to refine the prediction and consequently mitigate the accumulated errors in navigation.

Compared with conventional navigation models (Chen et al., 2021; 2022), **our prediction–update NeuroKalman architecture offers a unique advantage in preventing overfitting**. Existing models typically requires massive datasets to learn a generalized transition function, which are still prone to overfitting when data diversity is limited. In contrast, NeuroKalman retrieves relevant historical visual observations. This allows it to effectively utilize the necessary information to regularize VLN model training by enforcing temporal smoothness. In this way, we align the semantics of multiple local observations with global instructions, improving the next waypoint prediction. Experiments on TravelUAV benchmark (Wang et al., 2024b) show that NeuroKalman, fine-tuned on only 10% of the training data, significantly outperforms the initial model and the naive finetuning baselines. Our contributions can be summarized as follows:

- We point out the accumulation of error in continuous UAV navigation, and investigates a correction mechanism based on historical content and temporal smoothness for robust VLN model prediction & generalization.

- We formulate the navigation as a recursive Bayesian state estimation problem and propose the corresponding NeuroKalman framework. With the mathematical association between KDE of the likelihood function and the attention-based memory retrieval, we model the Bayesian prediction-update cycle and integrate the selected memory of historical observation to correct the prediction and mitigate state drift in continuous environments.

- With a randomly sampled subset of training data for model finetuning, our method clearly improve the model performance on the TravelUAV benchmark with a clear margin. This also verifies its effectiveness in mitigating overfitting when finetuning over limited data.

## 2. Related Works

**Vision-and-Language Navigation (VLN).** VLN research has evolved significantly from indoor environments to large-scale aerial scenarios (Anderson et al., 2018; Jain et al., 2019). Early approaches like AerialVLN (Liu et al., 2023) and CMA (Anderson et al., 2018) operated on discrete graphs, selecting actions from pre-defined steps (Krantz et al., 2020). Some methods rely on large-scale pre-training or data augmentation to improve generalization (Guhur et al., 2021; Chen et al., 2021). However, this setting is overly simplified and fails to match realistic UAV dynamics. Recent works, such as CityNav (Lee et al., 2025) and TravelUAV (Wang et al., 2024b), have shifted to continuous environments to better simulate real-world flights. OpenVLN (Lin et al., 2025) introduces a data-efficient approach for continuous control, while NavFoM (Zhang et al., 2025a) proposes a unified foundation model capable of processing multimodal inputs across varying horizons. Despite these advancements, existing methods typically rely on static networks with fixed weights. They lack the ability to correct accumulated errors during long flights, leading to the state drift problem.

**Temporal Context Modeling.** Temporal context modeling is crucial for long-horizon navigation, where the agent needs to incorporate historical context for trajectory planning to avoid getting lost. Early methods used recurrent networks like LSTMs (Anderson et al., 2018) or GRUs (Chung et al., 2014) to encode history, but they suffer from information loss over long horizons (Fried et al., 2018). Later works introduced explicit memory structures to extend the context window. For example, MapNet (Henriques & Vedaldi, 2018) and Transformer-XL (Dai et al., 2019) store historical states, while recent methods like SkyVLN (Li et al., 2025), OpenFly (Gao et al., 2025), and CityNavAgent (Zhang et al., 2025b) utilize topological maps or key-frames to assist reasoning. However, these approaches typically treat memory as a passive buffer, simply aggregating historical features with current observations. In contrast, we adopt a "retrieve-

to-correct" paradigm (Khandelwal et al., 2019; Borgeaud et al., 2022). Instead of simple feature concatenation, we use retrieved memory as probabilistic evidence to explicitly rectify the agent's belief state through Bayesian fusion, actively correcting potential drift.

**Deep Bayesian Filtering & State Estimation.** Addressing the distribution shift in deployment environments has recently popularized Test-Time Adaptation (TTA) techniques in embodied AI (Wang et al., 2020; Kumar et al., 2021). Recent methods like FEEDTTA (Kim et al.) and FSTTA (Gao et al., 2023) attempt to mitigate drift via feedback-based reinforcement learning or online gradient updates. However, in UAV-VLN, the lack of reliable supervision often causes these methods to reinforce existing errors (Niu et al., 2022). Alternatively, Deep Bayesian Filtering combines neural networks with probabilistic models (Kloss et al., 2021; Revach et al., 2022). While effective at learning transition prior dynamics, these approaches struggle to define a robust likelihood for high-dimensional visual inputs, often reverting to parametric models that are themselves prone to drift (Haarnoja et al., 2016; Becker et al., 2019). In contrast, NeuroKalman addresses this by framing navigation as Recursive Bayesian Estimation. Instead of updating model weights like TTA, we correct the belief state. By deriving the likelihood directly from memory retrieval, our framework offers a stable solution to state drift.

## 3. Method

To address the error accumulation inherent in dead-reckoning (Hong et al., 2021; Chen et al., 2021), we depart from the conventional paradigm that treats navigation purely as a sequential prediction task (Anderson et al., 2018; Fan et al., 2023), but instead reframe it as a recursive Bayesian state estimation problem. As illustrated in Figure 2, this formulation enables us to structurally decouple the *Prediction Block*, for initial prior estimation, from the *Update Block* for refinement from historical measurements, transforming navigation from a blind rollout into a reliable inference process. Accordingly, we posit that robust navigation relies on maintaining a probabilistic belief over the state space. This is achieved by fusing the motion dynamics prior with reliable measurements via the *Kalman Correction* to obtain a rectified posterior that prevents drift.

### 3.1. Problem Formulation: Navigation as Filtering

Formally, we consider a UAV operating in a continuous 3D environment (Krantz et al., 2020; Wang et al., 2024b). At each time step $t$, the model receives an observation tuple $o_t = \{v_t, p_t, l\}$, comprising multi-view visual inputs $v_t$ (*i.e.*, front, back, down, left, right), current 3D coordinates $p_t$, and the global natural language instruction $l$. The model predicts a next waypoint $w_t$ for UAV's execution. Central to our for-

mulation is the high-dimensional latent belief state $\mathbf{z}_t \in \mathbb{R}^d$. Unlike simple waypoint coordinates, $\mathbf{z}_t$ encodes a semantic understanding of the UAV's position and environmental context from time step $t$. Our core objective is to estimate the posterior distribution of this state, $P(\mathbf{z}_t|o_{1:t}, w_{1:t-1})$, given the entire history of observations and waypoints. Under the standard Markov assumption, this posterior estimation decomposes into a recursive prediction-update cycle, known as the Bayes filter (Särkkä & Svensson, 2023):

$$\underbrace{P(\mathbf{z}_t|o_{1:t}, w_{1:t-1})}_{\text{Posterior}} \propto \underbrace{P(o_t|\mathbf{z}_t)}_{\text{Likelihood}} \times \underbrace{P(\mathbf{z}_t|\mathbf{z}_{t-1}, w_{t-1})}_{\text{Prior}} \quad (1)$$

Since these distributions are intractable in high-dimensional visual spaces, we propose NeuroKalman to structurally instantiate this logic within a neural architecture (Kloss et al., 2021; Haarnoja et al., 2016). As detailed in the following sections, we decouple Eq. 1 into three learnable modules:

- Predictive Prior (Section 3.2): A RNN-based predictor that models the transition $P(\mathbf{z}_t|\mathbf{z}_{t-1}, w_{t-1})$, serving as the dead-reckoning mechanism.

- Measurement Likelihood (Section 3.3): An MLLM that encodes multi-modal inputs with historical memory to parameterize the likelihood $P(o_t|\mathbf{z}_t)$, providing retrieved evidence as corrective anchors.

- The Kalman Correction (Section 3.4): A gating mechanism that dynamically fuses the Prior and Likelihood to compute the rectified Posterior $P(\mathbf{z}_t|o_{1:t}, w_{1:t-1})$.

### 3.2. The Prediction Step: Predictive Prior

The Prediction step is to primarily estimate the *a prior* belief of the current state, functioning as an internal motion model, based solely on learned transition dynamics. This step effectively serving as a dead-reckoning mechanism for the Bayesian filter, and we design an RNN with Gated Recurrent Unit (GRU) to parameterize the transition distribution $P(\mathbf{z}_t|\mathbf{z}_{t-1}, w_{t-1})$.

Formally, at time step $t$, this module processes three inputs: the previous posterior $\mathbf{z}_{t-1}$, which serves as the rectified belief state implicitly encoding the UAV's optimal position at $t-1$; the previous waypoint $\mathbf{w}_{t-1}$, defined as the UAV's displacement vector in the body coordinate system from $t-1$ to $t$; and the hidden state $\mathbf{h}_{t-1}$, which encodes the historical motion dynamics accumulated from the initial step up to $t-1$. To initialize the recursive process at $t=0$, we set the initial state $\mathbf{z}_0$ to the measurement output of the MLLM at the initial position (denoted as $\mathbf{r}_0$). The initial hidden state $\mathbf{h}_0$ is then obtained by mapping $\mathbf{z}_0$ through a MLP layer with a Tanh activation to align with the GRU's latent space. Based on these inputs, the GRU updates its

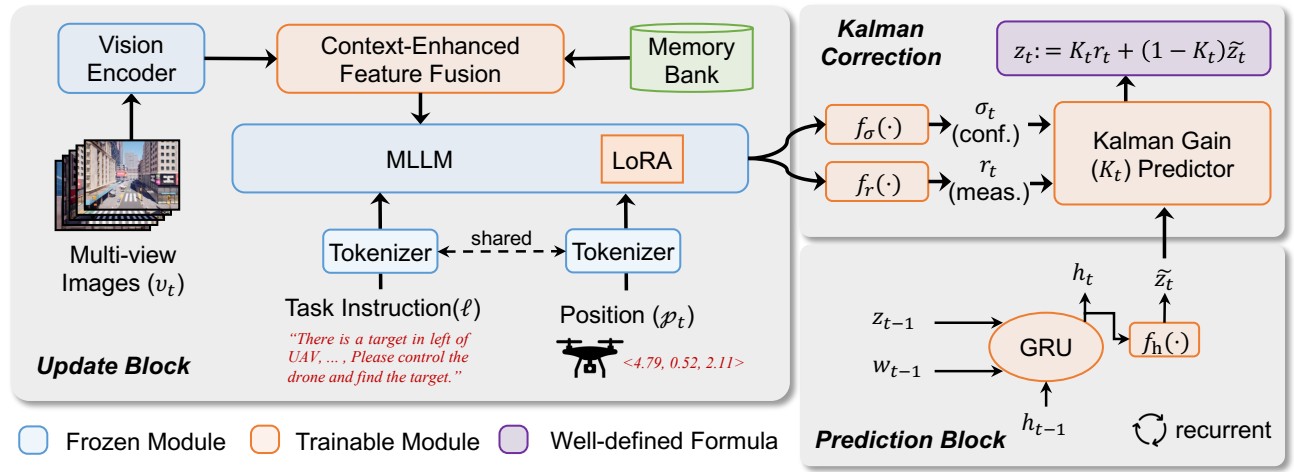

*Figure 2.* **NeuroKalman framework** aims to leverage temporal context to enhance next step prediction in navigation. Specifically, we follow the logic in classic Kalman filtering (Särkkä & Svensson, 2023), and consider the *Prediction* and *Update* steps (Kalman, 1960), *i.e.*, the former one makes initial estimation while the latter one estimates measurement representation $\mathbf{r}_t$ for core Kalman correction. In detail, the *Prediction Block* employs a GRU to roughly model the motion dynamics to predict the prior state $\tilde{\mathbf{z}}_t$ with updated hidden state $\mathbf{h}_t$, according to the posterior state $\mathbf{z}_{t-1}$ in the last step. Then, with the confidence scalar $\sigma_t$ predicted by the *Update Block*, the Kalman Gain $K_t$ is estimated on the representation space for correction. The waypoint prediction $\phi(\mathbf{z}_t)$ is omitted for the clarity of illustration while the variables $\mathbf{r}_t, \tilde{\mathbf{z}}_t$ can be both fed in $\phi(\cdot)$ for augmented supervision.

internal hidden state and projects the prior estimate $\tilde{\mathbf{z}}_t$ as follows:

$$\mathbf{h}_t = \text{GRU}([\mathbf{z}_{t-1}, \mathbf{w}_{t-1}], \mathbf{h}_{t-1}) \qquad (2)$$

$$\tilde{\mathbf{z}}_t = \text{MLP}_{prior}(\mathbf{h}_t) \qquad (3)$$

Here, $\tilde{\mathbf{z}}_t$ denotes the prior prediction st time step $t$. Crucially, this prediction is a "blind" process: it is derived purely from dead-reckoning logic without accessing the current visual observation $v_t$ (Banino et al., 2018). While the GRU effectively captures temporal dependencies and smooths the trajectory, it remains a purely parametric model governed by fixed weights. In unseen scenarios during training or data-scarce regimes, reliance on this motion dynamic prediction $\tilde{\mathbf{z}}_t$ inevitably leads to error accumulation, causing the latent belief to drift from the true manifold. This necessitates the subsequent likelihood correction step (Liu et al., 2023; Wang et al., 2024b).

### 3.3. The Update Step: Measurement Likelihood

As the Prediction step primarily uses motion dynamics for prior estimation, we implement the *Update Block* to fuse the *historical memory* functioning as external information, with the current observations and correspondingly implement a Multimodal Large Language Model (MLLM) to parameterize the measurement likelihood. At each time step $t$, we first augment the current visual input $v_t$ with relevant historical context retrieved from the episodic memory bank $\mathcal{M}$. The MLLM then jointly processes the memory-augmented visual features, the global instruction $l$, and the position of the UAV $p_t$ to output the latent measurement $\mathbf{r}_t$ along with a

confidence score $\sigma_t \in [0, 1]$ representing the measurement uncertainty. This $\mathbf{r}_t$ serves as the measurement likelihood for the subsequent Kalman correction.

**Memory Construction.** To facilitate the likelihood estimation, the memory bank $\mathcal{M}$ is constructed incrementally to store high-fidelity historical visual anchors. We adopt a post-correction storage strategy (Yang et al., 2024; Shi et al., 2025): the visual representation corresponding to the waypoint localization decoded from the *rectified posterior state* is appended to $\mathcal{M}$ only if the system's confidence score exceeds a reliability threshold (*i.e.*, $\sigma_t > 0.5$). Formally, the bank is defined as $\mathcal{M} = \{(\mathbf{k}_i, \mathbf{v}_i)\}_{i=1}^N$, where $\mathbf{k}_i = \mathbf{v}_i$ encodes the fixed visual features of selected snapshots. This selective storage mechanism ensures that the MLLM always attends to "validated anchors" (Chen et al., 2022), preventing the accumulation of noisy or ambiguous measurements.

**Retrieval as Kernel Density Estimation.** We rigorously formulate the memory retrieval process not merely as feature matching, but as Kernel Density Estimation (KDE) over the visual feature manifold. Given that our retrieval operates purely within the visual domain, where the query is the current visual feature $\mathbf{f}_t$ and the memory stores historical visual features $\{\mathbf{f}_i\}_{i=1}^N$, our goal is to derive a refined visual evidence $\hat{\mathbf{z}}_{evi}$. From a statistical perspective, we treat the historical features as empirical samples drawn from the underlying observation distribution. We employ the Nadaraya-Watson kernel regression estimator to approximate the expected canonical feature $\mathbb{E}[\mathbf{f}|o_t]$ on the visual

manifold (Nadaraya, 1964; Watson, 1964):

$$\hat{\mathbf{z}}_{evi} = \frac{\sum_{i=1}^{N} \mathcal{K}(\mathbf{f}_t, \mathbf{f}_i) \cdot \mathbf{f}_i}{\sum_{j=1}^{N} \mathcal{K}(\mathbf{f}_t, \mathbf{f}_j)} \qquad (4)$$

where $\mathbf{f}_t$ serves as the query, $\mathbf{f}_i$ serves as both key and value, and $\mathcal{K}(\cdot, \cdot)$ is a kernel function measuring visual similarity. In our NeuroKalman architecture, we utilize the scaled dot-product attention mechanism. Defining the kernel as $\mathcal{K}(\mathbf{x}, \mathbf{y}) = \exp(\frac{\mathbf{x}^\top \mathbf{y}}{\sqrt{d}})$, Eq. 4 becomes mathematically equivalent to the Softmax Attention operation (Katharopoulos et al., 2020; Choromanski et al., 2020):

$$\alpha_i = \text{Softmax}\left(\frac{\mathbf{f}_t^\top \mathbf{f}_i}{\sqrt{d}}\right), \quad \hat{\mathbf{z}}_{evi} = \sum_{i=1}^{N} \alpha_i \mathbf{f}_i \qquad (5)$$

Through this derivation, the retrieved vector $\hat{\mathbf{z}}_{evi}$ functions as a measurement correction. The term $\alpha_i$ represents the posterior probability $P(\mathbf{f}_i | \mathbf{f}_t)$ that the current visual observation belongs to the same local manifold as history $i$. Consequently, $\hat{\mathbf{z}}_{evi}$ aggregates historical anchors to mitigate high-frequency visual noise before state estimation. Unlike the GRU prior which propagates uncertainty blindly, $\hat{\mathbf{z}}_{evi}$ leverages observational evidence to stabilize the likelihood term, ensuring robustness in out-of-distribution scenarios.

### 3.4. The Kalman Correction: Fusion as Gain

The final step fuses the potentially drifting prior estimate $\tilde{\mathbf{z}}_t$ (from Section 3.2) with the encoded measurement $\mathbf{r}_t$ (from Section 3.3) to obtain the final corrected posterior $\mathbf{z}_t$. We design a fusion mechanism that structurally mirrors the classic Kalman Filter update equation (Kalman, 1960):

$$\mathbf{z}_{post} = \mathbf{z}_{prior} + \mathbf{K}_t(\mathbf{y}_t - \mathbf{H}\mathbf{z}_{prior}) \qquad (6)$$

In our NeuroKalman framework, the measurement $\mathbf{y}_t$ corresponds to the encoded measurement $\mathbf{r}_t$. $\mathbf{z}_{prior}$ and $\mathbf{z}_{post}$ correspond to the prior estimate $\tilde{\mathbf{z}}_t$ and final corrected posterior $\mathbf{z}_t$, respectively. We assume an identity measurement matrix $\mathbf{H} = \mathbf{I}$ since both the prior and measurement are mapped into the same aligned latent feature space (Haarnoja et al., 2016).

**Learnable Kalman Gain.** The core of this correction is the optimal weighting between prior and measurement. Considering the intractability of explicitly estimating noise covariances (*i.e.,* $\mathbf{Q}$ and $\mathbf{R}$) of traditional Kalman Filtering in high-dimensional latent space, we formulate a learnable gating network to approximate the Kalman Gain $\mathbf{K}_t$ (Revach et al., 2022). It dynamically computes the element-wise uncertainty based on the current context:

$$\mathbf{K}_t = \text{Sigmoid}\left(\mathbf{W}_g[(\mathbf{r}_t - \tilde{\mathbf{z}}_t) ; \phi(\sigma_t)] + \mathbf{b}_g\right) \qquad (7)$$

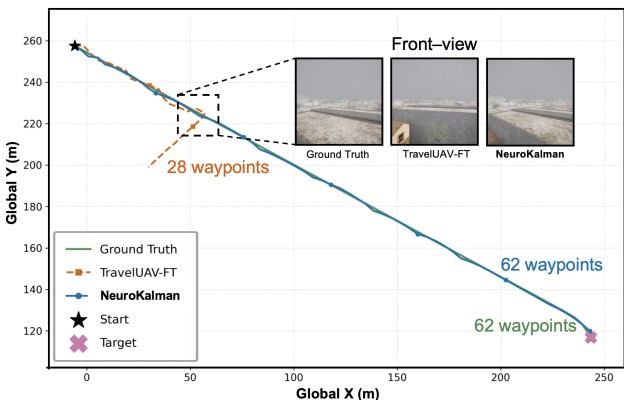

*Figure 3.* **Demonstration of trajectory rectification.** The TravelUAV-FT relies solely on parametric predictions to estimate its trajectory, resulting in obvious trajectory drift. NeuroKalman rectifies its position by integrating Kalman correction.

where $[\cdot ; \cdot]$ denotes concatenation, $\phi(\cdot)$ is a learnable MLP projection mapping the confidence score $\sigma_t$ to the feature dimension, and $\text{Sigmoid}(\cdot)$ is the activation function ensuring the gain $\mathbf{K}_t \in (0, 1)^d$.

**Bayesian Update.** Using the computed gain, the system updates the belief state. The fusion equation is defined as:

$$\mathbf{z}_t = (1 - \mathbf{K}_t) \odot \tilde{\mathbf{z}}_t + \mathbf{K}_t \odot \mathbf{r}_t \qquad (8)$$

$$= \tilde{\mathbf{z}}_t + \mathbf{K}_t \odot (\mathbf{r}_t - \tilde{\mathbf{z}}_t) \qquad (9)$$

Equation 9 is algebraically identical to the standard Kalman correction form (Eq. 6), where $(\mathbf{r}_t - \tilde{\mathbf{z}}_t)$ represents the *residual*—the difference between the external measurement and the predicted prior.

**Dynamic Uncertainty Regulation.** The learnable Kalman Gain $\mathbf{K}_t$ functions as an adaptive regulator. When the measurement is confident, the system heavily weighs the innovation to pull the state towards the validated measurement. Conversely, when the measurement is ambiguous, it relies more on the smooth internal dynamics of the Prior. This mechanism ensures that the final posterior $\mathbf{z}_t$ remains anchored to the true trajectory manifold, effectively cancelling the state drift. Finally, $\mathbf{z}_t$ is passed to the waypoint predictor to predict the next waypoint and fed back into the next prediction step as $\mathbf{z}_{t-1}$. As visualized in Figure 3, this correction mechanism actively performs micro-adjustments to re-align the belief state with the ground truth, effectively preventing the trajectory drift characteristic of the uncorrected baseline (TravelUAV-FT).

## 4. Experiments

In this section, we empirically validate the effectiveness of NeuroKalman on the challenging TravelUAV benchmark, with a specific focus on data efficiency and generalization. We first describe the experimental setup and evaluation pro-

*Table 1.* Experimental results on the UAV-Need-Help test seen set, grouped by assistant levels (L1-L3). **Bold** indicates the best performance among all methods within each assistant level.

| Assistant | Methods | Full | | | | Easy | | | | Hard | | | |
|---|---|---|---|---|---|---|---|---|---|---|---|---|---|
| | | NE ↓ | SR ↑ | OSR ↑ | SPL ↑ | NE ↓ | SR ↑ | OSR ↑ | SPL ↑ | NE ↓ | SR ↑ | OSR ↑ | SPL ↑ |
| L1 | Random Action | 222.20 | 0.14 | 0.21 | 0.07 | 142.07 | 0.26 | 0.39 | 0.13 | 320.12 | 0.00 | 0.00 | 0.00 |
| | Fixed Action | 188.61 | 2.27 | 8.16 | 1.40 | 121.36 | 3.48 | 11.48 | 2.14 | 270.69 | 0.79 | 4.09 | 0.49 |
| | CMA (Anderson et al., 2018) | 135.73 | 8.37 | 18.72 | 7.90 | 84.89 | 11.48 | 24.52 | 10.68 | 197.77 | 4.57 | 11.65 | 4.51 |
| | TravelUAV (Wang et al., 2024b) | 106.28 | 16.10 | 44.26 | 14.30 | 68.78 | 18.84 | 47.61 | 16.39 | 152.04 | 12.76 | 40.16 | 11.76 |
| | TravelUAV-FT (Wang et al., 2024b) | 99.79 | 17.56 | 41.89 | 14.71 | 64.10 | 20.69 | 45.98 | 16.79 | 143.85 | 13.70 | 36.85 | 12.15 |
| | OpenVLN (Lin et al., 2025) | 125.97 | 14.39 | 28.03 | 12.94 | 87.96 | 15.22 | 30.64 | 13.31 | 175.54 | 13.32 | 24.62 | 12.55 |
| | **NeuroKalman (Ours)** | **71.56** | **25.86** | **58.73** | **22.43** | **42.70** | **30.52** | **62.70** | **25.86** | **105.07** | **20.11** | **53.90** | **18.21** |
| L2 | CMA (Anderson et al., 2018) | 141.55 | 7.02 | 15.39 | 6.54 | 87.77 | 9.55 | 19.87 | 8.74 | 207.18 | 3.94 | 9.92 | 3.94 |
| | TravelUAV (Wang et al., 2024b) | 120.57 | 12.98 | 37.38 | 11.30 | 76.89 | 17.55 | 43.48 | 15.01 | 186.22 | 7.40 | 29.92 | 6.76 |
| | TravelUAV-FT (Wang et al., 2024b) | 112.20 | 14.05 | 35.50 | 11.70 | 72.50 | 18.80 | 42.00 | 15.50 | 178.00 | 8.20 | 27.50 | 7.50 |
| | OpenVLN (Lin et al., 2025) | 129.68 | 13.83 | 25.97 | 12.18 | 82.80 | 17.30 | 31.41 | 14.70 | 178.32 | 10.24 | 20.31 | 9.57 |
| | **NeuroKalman (Ours)** | **87.11** | **22.32** | **53.68** | **19.40** | **46.50** | **27.60** | **59.50** | **23.00** | **138.00** | **15.80** | **46.50** | **14.50** |
| L3 | CMA (Anderson et al., 2018) | 140.93 | 4.89 | 11.56 | 4.41 | 83.58 | 7.35 | 17.81 | 6.53 | 210.91 | 1.89 | 3.94 | 1.83 |
| | TravelUAV (Wang et al., 2024b) | 146.32 | 6.31 | 15.39 | 5.10 | 93.15 | 9.55 | 21.94 | 7.32 | 215.85 | 2.36 | 7.40 | 2.17 |
| | TravelUAV-FT (Wang et al., 2024b) | 135.00 | 4.51 | 12.06 | 3.70 | 89.00 | 6.00 | 17.11 | 4.70 | 198.78 | 2.68 | 5.83 | 2.47 |
| | OpenVLN (Lin et al., 2025) | 146.42 | 2.70 | 6.96 | 2.19 | 92.90 | 4.86 | 11.38 | 3.89 | 201.96 | 0.47 | 2.36 | 0.43 |
| | **NeuroKalman (Ours)** | **107.55** | **8.89** | **17.28** | **7.01** | **83.00** | **11.24** | **23.63** | **10.65** | **179.07** | **5.98** | **9.45** | **4.89** |

tocol, then present main results, ablation studies, and a detailed analysis of state drift problem.

## 4.1. Experimental Setup

We evaluate our framework on the TravelUAV benchmark (Wang et al., 2024b), focusing on robust performance under data constraints. We detail the dataset, baselines, metrics, and our specific implementation protocols below.

**Dataset & Simulation Environment.** We adopt the TravelUAV benchmark and its UAV-Need-Help dataset (Wang et al., 2024b), simulated in the high-fidelity AirSim environment (Shah et al., 2017) which provides realistic physics and diverse outdoor conditions (urban, snowy, meadow, etc). The dataset contains 12,149 human-operated trajectories annotated with 89 object categories. Following the official protocol, the data is split as follows: the Training set contains 9,152 trajectories across 20 scenes; the Test-Seen set has 1,410 trajectories from training scenes; the Test-Unseen-Map set comprises 958 trajectories from 2 entirely novel scenes; and the Test-Unseen-Object set contains 629 trajectories targeting novel objects. Trajectories are categorized by length: "Easy" ($< 250$m) and "Hard" ($\geq 250$m), with target distances ranging from 50 to 400 meters.

**Baselines.** We compare NeuroKalman against a comprehensive set of baselines: (1) Random/Fixed Action: Naive models that select random poses or map instructions to fixed movements; (2) CMA (Anderson et al., 2018): A standard bi-directional LSTM model using cross-modal attention; (3) TravelUAV baseline (Wang et al., 2024b): The official strong baseline provided by the benchmark; (4) OpenVLN (Lin et al., 2025) and NavFoM (Zhang et al., 2025a): State-of-the-art approaches for continuous UAV navigation

and multimodal foundation models. Note that for the TravelUAV baseline, we report results under two settings: trained on full data (denoted as TravelUAV) and fine-tuned on 10% training data (denoted as TravelUAV-FT).

**Metrics.** We report four standard metrics (Anderson et al., 2018): Navigation Error (NE), the average distance between the UAV's final position and the target; Success Rate (SR), the percentage of episodes where the agent stops within 20 meters of the target; Oracle Success Rate (OSR), the success rate by accounting for the minimum distance to the target achieved at any point during the navigation trajectory; and Success weighted by Path Length (SPL), which balances success with trajectory efficiency.

**Implementation Details.** We implement our model using PyTorch on 4×NVIDIA RTX A6000 GPUs. We freeze the MLLM backbone (EVA-CLIP (Sun et al., 2023) as the visual encoder and Vicuna-7B (Chiang et al., 2023) as the language backbone) and only compute gradients for the visual projector, waypoint predictor, and LoRA (Hu et al., 2022) layers using the Adam optimizer (Learning Rate=$5e-5$, Batch Size=16). We also use an additional L1 loss to simultaneously supervise both the predictive prior ($\tilde{z}_t$) and measurement ($r_t$), with a coefficient factor of 0.2. We adopt limited data fine-tuning for robustness evaluation. Specifically, both NeuroKalman and the TravelUAV are initialized with weights pre-trained on the full data, yet are subsequently fine-tuned using only a fixed random 10% subset of the training data.

## 4.2. Main Results

We conduct a comprehensive evaluation of NeuroKalman on the TravelUAV benchmark. Accordingly, we compare our

*Table 2.* Experimental results with the L1 assistant on the UAV-Need-Help test unseen set. **Bold** indicates the best results.

| Split | Methods | Full | | | | Easy | | | | Hard | | | |
|---|---|---|---|---|---|---|---|---|---|---|---|---|---|
| | | NE ↓ | SR ↑ | OSR ↑ | SPL ↑ | NE ↓ | SR ↑ | OSR ↑ | SPL ↑ | NE ↓ | SR ↑ | OSR ↑ | SPL ↑ |
| UO | Random Action | 260.14 | 0.16 | 0.16 | 0.16 | 174.10 | 0.48 | 0.48 | 0.48 | 302.96 | 0.00 | 0.00 | 0.00 |
| | Fixed Action | 212.84 | 3.66 | 9.54 | 2.16 | 151.66 | 6.70 | 13.88 | 3.72 | 243.29 | 2.14 | 7.38 | 1.38 |
| | CMA (Anderson et al., 2018) | 155.79 | 9.06 | 16.06 | 8.68 | 102.92 | 14.83 | 22.49 | 13.90 | 182.09 | 6.19 | 12.86 | 6.08 |
| | TravelUAV (Wang et al., 2024b) | 118.11 | 22.42 | 46.90 | 20.51 | 86.12 | 24.40 | 49.28 | 22.03 | 134.03 | 21.43 | 45.71 | 19.75 |
| | TravelUAV-FT (Wang et al., 2024b) | 112.01 | 23.53 | 42.13 | 20.29 | 64.80 | 34.45 | 54.07 | 28.97 | 135.51 | 18.10 | 36.19 | 15.97 |
| | NavFoM (Zhang et al., 2025a) | 108.04 | 29.83 | 47.99 | 27.20 | 70.51 | 32.54 | 50.72 | 29.54 | 133.01 | **28.03** | 46.18 | **25.64** |
| | **NeuroKalman (Ours)** | **71.01** | **32.48** | **60.82** | **28.50** | **44.50** | **42.50** | **66.50** | **37.37** | **84.50** | 27.50 | **58.00** | 24.50 |
| UM | Random Action | 202.98 | 0.00 | 0.00 | 0.00 | 158.46 | 0.00 | 0.00 | 0.00 | 265.88 | 0.00 | 0.00 | 0.00 |
| | Fixed Action | 180.47 | 0.52 | 2.61 | 0.39 | 132.89 | 0.89 | 4.28 | 0.67 | 247.72 | 0.00 | 0.25 | 0.00 |
| | CMA (Anderson et al., 2018) | 141.68 | 2.30 | 10.02 | 2.16 | 102.29 | 3.57 | 14.26 | 3.33 | 197.35 | 0.50 | 4.03 | 0.50 |
| | TravelUAV (Wang et al., 2024b) | 138.80 | 4.18 | 20.77 | 3.84 | 102.94 | 4.63 | 22.82 | 4.24 | 189.46 | 3.53 | 17.88 | 3.28 |
| | TravelUAV-FT (Wang et al., 2024b) | 117.84 | 4.68 | 19.03 | 3.17 | 87.50 | 5.13 | 21.39 | 5.06 | 160.79 | 4.05 | 15.69 | 3.32 |
| | NavFoM (Zhang et al., 2025a) | 125.10 | 6.30 | 18.95 | 5.68 | 102.41 | 6.77 | 20.07 | 6.04 | 170.58 | 5.36 | 15.71 | 4.97 |
| | **NeuroKalman (Ours)** | **100.32** | **8.34** | **34.15** | **7.12** | **69.50** | **9.15** | **38.50** | **7.50** | **140.00** | **7.20** | **28.00** | **6.50** |

*Table 3.* Low-data results on L1 Test-Seen. NeuroKalman reports mean±std over three 10% trajectory-level random subsets. **Bold** indicates the best results.

| Methods | Protocol | NE ↓ | SR ↑ | OSR ↑ | SPL ↑ |
|---|---|---|---|---|---|
| TravelUAV (Wang et al., 2024b) | 100% data | 106.28 | 16.10 | 44.26 | 14.30 |
| TravelUAV (Wang et al., 2024b) | 10% w/o full pretrain | 122.63 | 13.19 | 37.16 | 11.71 |
| **NeuroKalman (Ours)** | **10% w/o full pretrain** | **85.89±2.71** | **22.70±0.87** | **45.04±1.09** | **19.04±0.70** |

method against several state-of-the-art baselines across two aspects: (1) Data Efficiency on the Test-Seen split, underscoring the model's robustness in long-horizon navigation; and (2) Generalization on the Test-Unseen splits (across both maps and objects), highlighting the model's capability to handle distribution shifts in unknown scenarios.

**Data Efficiency under Limited Fine-Tuning.** Table 1 summarizes the Test-Seen results under the limited fine-tuning protocol. Specifically, NeuroKalman and TravelUAV-FT are initialized from full-data pretrained weights and then fine-tuned on the same fixed 10% subset of the training data. Therefore, this comparison evaluates how effectively each model adapts with limited fine-tuning data. As shown, NeuroKalman achieves a significant performance lead across all metrics. For example, in the L1 Full split, our method attains a SR of 25.86%, surpassing TravelUAV-FT (17.56%) by a clear margin. This advantage is even more pronounced in the Hard split (> 250m), where we improve the OSR from 36.85% (TravelUAV-FT) to 53.90% (Ours). This indicates that while the parametric TravelUAV-FT exhibits signs of overfitting and struggles to maintain trajectory consistency when starved of data, our method effectively mitigates this by integrating Kalman correction. **By anchoring the internal belief state to retrieved historical evidence, NeuroKalman prevents the progressive error accumulation.** Furthermore, our model consistently delivers superior performance when compared to TravelUAV. Notably, despite using only 10% of the training data, NeuroKalman outperforms the TravelUAV on the L1 Hard split, reducing the NE

from 152.04 to 105.07. This further demonstrates the robustness of NeuroKalman, validating that decoupling prediction and correction enables precise navigation.

**Data Efficiency without Full-Data Pretraining.** The above limited fine-tuning protocol initializes both NeuroKalman and TravelUAV from weights pre-trained on the full training set. To verify that the improvement is not merely inherited from full-data pretraining, we further train the model without full-data pretraining using only 10% of TravelUAV training trajectories. The subset is uniformly sampled at the trajectory level with fixed random seeds, and the sampled splits preserve the Easy/Hard distribution of the full set (51.8%/48.2% on average vs. 52.4%/47.6% in the full training set).

As shown in Table 3, NeuroKalman substantially outperforms TravelUAV under the 10% low-data setting without full-data pretraining, reducing NE by 30.0% and improving SR by 72.1%. It also remains competitive with, and in most metrics surpasses, the fully trained TravelUAV baseline, despite using only 10% training trajectories. Meanwhile, TravelUAV only predicts waypoints step-by-step from current observations without explicit long-horizon error correction, leaving its performance heavily bottlenecked by the MLLM's inherent capacity and resulting in similarly low performance under both 10% and 100% data. This suggests that **NeuroKalman breaks this bottleneck via explicit error correction and temporal modeling, demonstrating its low-data efficiency.**

*Table 4.* Effectiveness of fusion mechanism. All methods are implemented with the L1 assistant on the UAV-Need-Help test seen set. **Bold** indicates the best results among all methods.

| Methods | Full | | | | Easy | | | | Hard | | | |
|---|---|---|---|---|---|---|---|---|---|---|---|---|
| | NE↓ | SR↑ | OSR↑ | SPL↑ | NE↓ | SR↑ | OSR↑ | SPL↑ | NE↓ | SR↑ | OSR↑ | SPL↑ |
| $\mathbf{K}_t = 0.1$ | 217.09 | 0.00 | 0.00 | 0.00 | 132.81 | 0.00 | 0.00 | 0.00 | 302.56 | 0.00 | 0.00 | 0.00 |
| $\mathbf{K}_t = 0.5$ | 83.14 | 24.12 | 53.74 | 19.40 | 46.50 | 27.60 | 59.51 | 23.00 | 138.00 | 19.84 | 46.61 | 17.30 |
| $\mathbf{K}_t = 0.9$ | 100.96 | 18.05 | 44.15 | 15.35 | 58.18 | 19.92 | 52.11 | 18.43 | 154.50 | 15.80 | 34.33 | 14.50 |
| **Learnable** | **71.56** | **25.86** | **58.73** | **22.43** | **42.70** | **30.52** | **62.70** | **25.86** | **105.07** | **20.11** | **53.90** | **18.21** |

*Table 5.* Impact of memory history length. All methods are implemented with the L1 assistant on the UAV-Need-Help test seen set. **Bold** indicates the best results among all methods.

| Methods | Full | | | | Easy | | | | Hard | | | |
|---|---|---|---|---|---|---|---|---|---|---|---|---|
| | NE↓ | SR↑ | OSR↑ | SPL↑ | NE↓ | SR↑ | OSR↑ | SPL↑ | NE↓ | SR↑ | OSR↑ | SPL↑ |
| $M = 5$ | 84.39 | 21.23 | 53.46 | 18.18 | 51.56 | 25.03 | 56.96 | 21.21 | 125.00 | 16.54 | 49.13 | 15.09 |
| $M = 10$ | **71.56** | **25.86** | **58.73** | **22.43** | **42.70** | **30.52** | **62.70** | **25.86** | **105.07** | **20.11** | **53.90** | **18.21** |
| $M = 15$ | 77.17 | 23.77 | 56.42 | 20.16 | 47.75 | 28.10 | 60.15 | 23.33 | 116.21 | 18.43 | 51.81 | 17.39 |

**Generalization on the Test-Unseen Split.** We further evaluate the model's generalization by testing on the Test-Unseen-Object and Test-Unseen-Map splits. As summarized in Table 2, NeuroKalman consistently outperforms baseline methods (Anderson et al., 2018; Wang et al., 2024b) in these novel settings on the Full split. For example, on the Unseen Objects (UO) split, our method achieves a SR of 32.48% and an OSR of 60.82%, significantly surpassing the strong NavFoM baseline (SR: 29.83%, OSR: 47.99%). The performance gap is even more evident in the challenging Unseen Maps (UM) split, where NeuroKalman nearly doubles the SR of TravelUAV (8.34% vs. 4.18%). While NavFoM achieves a slight advantage in SR and SPL on the Hard split of unseen objects due to its large-scale data pretraining, NeuroKalman maintains a considerably lower NE (84.50 vs. 133.01) and a higher OSR (58.00% vs. 46.18%). These results highlight the limitation of purely parametric baselines, which suffer from performance degradation due to error accumulation caused by state drift when facing unknown topologies or objects. In contrast, the generalization of NeuroKalman validates the efficacy of Bayesian fusion mechanism. **By adaptively regulating the reliance between the GRU's kinematic prior and the measurement likelihood, our framework successfully utilizes retrieval-based anchors to correct accumulated errors.**

### 4.3. Ablation Study

**Effectiveness of Fusion Mechanism.** Next, we investigate whether a learnable Kalman Gain is superior to fixed scalar values in balancing the motion prior and measurement evidence. As presented in Table 4, fixed strategies fail to achieve optimal navigation results. A strong bias towards the Prior ($\mathbf{K}_t = 0.1$) leads to catastrophic failure. Conversely, relying heavily on the Measurement ($\mathbf{K}_t = 0.9$)

also yields suboptimal performance, with the SR dropping to 18.05% on the Full split compared to the balanced setting. Even the best fixed strategy ($\mathbf{K}_t = 0.5$) is inferior to our approach. In contrast, our learnable gating mechanism consistently achieves superior performance across all metrics, reaching the highest SR of 25.86% and the lowest NE of 71.56. We think that the failure at $\mathbf{K}_t = 0.1$ confirms that without sufficient external correction, the model suffers from unbounded state drift. Meanwhile, the degradation at $\mathbf{K}_t = 0.9$ suggests that ignoring temporal smoothness makes the model vulnerable to noisy retrievals. Our learnable gating network effectively acts as an uncertainty switch, which dynamically **integrates internal dynamics and external measurements to maintain robust estimation across diverse noise regimes.**

**Impact of Memory History Length.** We also investigate the impact of memory history length $M$ to determine the optimal temporal context for our retrieval mechanism. As shown in Table 5, deviations from our design choice lead to distinct performance drops. A short history ($M = 5$) yields suboptimal results, with a higher NE of 84.39 on the Full split. Interestingly, extending the history excessively ($M = 15$) does not help but rather degrades performance, increasing the NE to 77.17. Our setting ($M = 10$) achieves the best performance with the lowest NE of 71.56. We think that a limited temporal window lacks sufficient historical anchors to recover from accumulated drift, whereas an excessive length introduces outdated visual features that act as noise, distracting the attention mechanism from relevant evidence. This indicates that **appropriate memory length ensures sufficient context for effective re-localization without introducing spurious correlations.**

**Additional Robustness Checks.** We further evaluate NeuroKalman's robustness to output-space smoothing and mem-

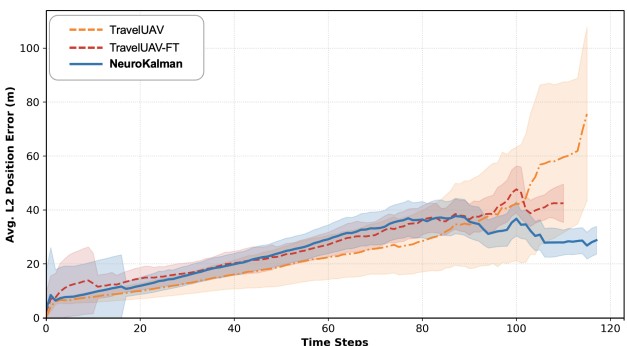

*Figure 4.* **Visualization of $L_2$ position error over time.** The baselines (orange and red dashed lines) show a continuous error increase on long trajectories. Conversely, NeuroKalman (blue solid line) keeps the error stable and prevents it from growing rapidly via effective Kalman correction.

*Table 6.* Comparison with post-hoc Kalman filtering on L1 Test-Seen set. **Bold** indicates the best results.

| Methods | NE ↓ | SR ↑ | OSR ↑ | SPL ↑ |
|---|---|---|---|---|
| TravelUAV | 106.28 | 16.10 | 44.26 | 14.30 |
| TravelUAV + Post-KF | 96.67 | 18.17 | 33.65 | 15.64 |
| **NeuroKalman (Ours)** | **71.56** | **25.86** | **58.73** | **22.43** |

ory noise by introducing a post-hoc Kalman filtering baseline and varying the memory insertion threshold. On the one hand, we compare against a post-hoc Kalman baseline that applies a constant-velocity Kalman filter to TravelUAV's predicted waypoints. As shown in Table 6, this output-space smoothing improves TravelUAV only mildly on L1 Test-Seen (NE/SR: 106.28/16.10 → 96.67/18.17), still far behind NeuroKalman (71.56/25.86), showing that **latent-space correction with semantic memory cannot be replaced by geometric smoothing alone.** On the other hand, we vary the memory insertion threshold $\sigma_t \in \{0.3, 0.5, 0.7, 0.9\}$. As shown in Table 7, NeuroKalman remains stable for $\sigma_t \geq 0.5$, with $\sigma_t = 0.5$ yielding the best performance. A low threshold ($\sigma_t = 0.3$) hurts performance by admitting noisy anchors. Our method exhibits a robust threshold range: memory is inserted only through post-correction selective storage, and the posterior is filtered via confidence-aware Kalman fusion, ensuring that noisy retrievals are not unconditionally trusted.

### 4.4. Drift Analysis

Finally, we analyze the state drift problem by visualizing the average $L_2$ position error over time in Figure 4. Intuitively, the baseline curves (TravelUAV and TravelUAV-FT) terminate earlier than NeuroKalman because baseline models are more prone to premature collisions or severe drift on long trajectories. In detail, their localization errors accumulate

*Table 7.* Sensitivity to memory insertion threshold on L1 Test-Seen set. **Bold** indicates the best results.

| Threshold | NE ↓ | SR ↑ | OSR ↑ | SPL ↑ |
|---|---|---|---|---|
| $\sigma_t = 0.3$ | 82.45 | 20.50 | 48.15 | 17.22 |
| $\sigma_t = 0.5$ | **71.56** | **25.86** | **58.73** | **22.43** |
| $\sigma_t = 0.7$ | 73.88 | 24.55 | 55.62 | 22.10 |
| $\sigma_t = 0.9$ | 75.45 | 24.18 | 53.30 | 21.75 |

over time and become especially evident after 100 steps. In contrast, the error of NeuroKalman stops growing and stays stable (around 30–40 meters) after a small initial rise. This demonstrates that our Kalman correction mechanism successfully **leverages retrieved memory anchors to periodically rectify the internal belief**, ultimately constraining error accumulation in long-horizon navigation.

## 5. Conclusion

In this work, we reframe the continuous navigation task as a Recursive Bayesian State Estimation problem to address the fundamental challenge of state drift inherent in open-loop parametric inference. We propose NeuroKalman, a framework that decouples navigation into prior prediction via motion dynamics and likelihood correction via memory anchors, while mathematically establishing the equivalence between attention-based retrieval and Kernel Density Estimation (KDE). Comprehensive experiments on TravelUAV benchmark demonstrate that our method achieves remarkable data efficiency and generalization, clearly outperforming strong baselines and regulating drift accumulation.

**Limitation.** The current implementation of the prediction prior utilizes a GRU-based RNN as baseline, which may be subject to information decay over exceptionally long horizons. However, the primary contribution of NeuroKalman is recursive Bayesian correction mechanism and we can easily apply another architecture to enhance the overall robustness.

## Acknowledgement

This work was supported in part by the National Natural Science Foundation of China (NSFC) Young Scientists Fund (Type B) under Grant No. 62522220; in part by the NSFC General Program under Grant No. 62172439, and in part by the NSFC General Program under Grant No. 62502405.

## Impact Statement

This work advances robust UAV navigation in GPS-denied environments, with potential for search-and-rescue and disaster relief. Although effective, rigorous verification processes and safety protocols are strictly necessary prior to integrating it into real-world embodied systems.

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

# A. Appendix

This supplementary material provides a comprehensive analysis of the NeuroKalman framework, substantiating its effectiveness through theoretical rigor, architectural ablation, and qualitative visualization. First, we formalize the navigation error dynamics, providing a mathematical proof of how the Kalman correction mechanism guarantees error contraction and serves as an implicit anchor regularization. Second, we present extended ablation studies to empirically validate the necessity of Bayesian structural decoupling and extra supervision signals. Finally, we offer intuitive visualizations of trajectory rectification, corroborating the model's ability to correct accumulated drift in long-horizon scenarios.

## A.1. Theoretical Justification

In this section, we provide a theoretical analysis of why the NeuroKalman framework effectively mitigates navigational drift and achieves high sample efficiency, even when fine-tuned on limited data. We analyze the error dynamics of the fusion mechanism and interpret the retrieval process as an implicit regularization.

### A.1.1. DRIFT CANCELLATION VIA ERROR CONTRACTION

Navigation drift can be formalized as the accumulation of error in the latent space over time. Let $\mathbf{z}_t^*$ be the ground-truth state on the manifold, and let $\epsilon_t = \|\mathbf{z}_t - \mathbf{z}_t^*\|$ be the estimation error at step $t$. In a standard recurrent baseline (*e.g.,* vanilla GRU), the error dynamics are governed by the Lipschitz constant (Miller & Hardt, 2018) of the transition function $\lambda_{gru}$:

$$\epsilon_t^{gru} \leq \lambda_{gru} \cdot \epsilon_{t-1} + \delta_{noise} \tag{10}$$

Since navigation environments are complex, locally expanding dynamics typically imply $\lambda_{gru} > 1$. Consequently, the error grows exponentially with time ($\epsilon_t \propto \lambda_{gru}^t$), leading to inevitable drift. In our framework, the Kalman correction modifies these dynamics. Substituting the fusion Eq. 11 into the error term:

$$\mathbf{z}_t = (1 - \mathbf{K}_t) \odot \tilde{\mathbf{z}}_t + \mathbf{K}_t \odot \mathbf{r}_t \tag{11}$$

and assuming the encoded measurement $\mathbf{r}_t$ serves as a bounded approximation of the ground truth ($\|\mathbf{r}_t - \mathbf{z}_t^*\| \leq \xi$), the posterior error becomes:

$$\epsilon_t^{kalman} = \|(\mathbf{I} - \mathbf{K}_t)(\tilde{\mathbf{z}}_t - \mathbf{z}_t^*) + \mathbf{K}_t(\mathbf{r}_t - \mathbf{z}_t^*)\| \tag{12}$$

Applying the triangle inequality:

$$\epsilon_t^{kalman} \leq \|\mathbf{I} - \mathbf{K}_t\| \cdot \underbrace{\lambda_{gru}\epsilon_{t-1}}_{\text{Prior Error}} + \|\mathbf{K}_t\| \cdot \underbrace{\xi}_{\text{Retrieval Noise}} \tag{13}$$

Here, the term $(\mathbf{I} - \mathbf{K}_t)$ acts as a contraction matrix. As long as the gating network learns to assign non-zero gain ($\mathbf{K}_t > 0$) when the prior uncertainty is high, the spectral radius $\rho(\mathbf{I} - \mathbf{K}_t)$ becomes strictly less than 1. Thus, the Kalman Gain actively dampens the propagation of historical error. Even if the GRU prior drifts ($\lambda_{gru} > 1$), the fusion mechanism ensures the error remains bounded, technically proving the *drift cancellation* property.

### A.1.2. IMPLICIT ANCHOR REGULARIZATION

Why does the model generalize well with only 10% fine-tuning data? We further argue that the fusion mechanism introduces an inductive bias equivalent to *Anchor Regularization*. The classic Kalman update can be derived as the analytical solution to the following optimization problem (Bishop & Nasrabadi, 2006):

$$\mathbf{z}_t = \underset{\mathbf{z}}{\arg\min} \left( \underbrace{\frac{1}{2}\|\mathbf{z} - \tilde{\mathbf{z}}_t\|_{\Sigma_{prior}^{-1}}^2}_{\text{Trust Prediction}} + \underbrace{\frac{1}{2}\|\mathbf{z} - \mathbf{r}_t\|_{\Sigma_{obs}^{-1}}^2}_{\text{Trust Evidence}} \right) \tag{14}$$

This formulation reveals that incorporating $\mathbf{r}_t$ is mathematically equivalent to adding a regularization term to the loss landscape. The retrieved state $\mathbf{r}_t$ acts as a high-level semantic anchor, penalizing the model whenever the optimization variable $\mathbf{z}$ deviates from the evidence. During backpropagation, this anchor prevents the GRU parameters $\theta$ from overfitting to spurious trajectories. Instead of forcing $\theta$ to memorize the entire global map (which requires massive data), the parameters only need to learn the local smoothing function between anchors (Savinov et al., 2018; Guu et al., 2020). The global consistency is offloaded to the memory bank $\mathcal{M}$, thereby significantly reducing the sample complexity required to learn $\theta$.

*Table 8.* Experimental results of structural decoupling architecture. All experiments are conducted using the L1 assistant setting.

| Methods | Full | | | | Easy | | | | Hard | | | |
|---|---|---|---|---|---|---|---|---|---|---|---|---|
| | NE↓ | SR↑ | OSR↑ | SPL↑ | NE↓ | SR↑ | OSR↑ | SPL↑ | NE↓ | SR↑ | OSR↑ | SPL↑ |
| *Test Seen Set* | | | | | | | | | | | | |
| MBGRU | 75.35 | 24.68 | 54.87 | 20.43 | 48.80 | 29.37 | 60.41 | 23.66 | 108.05 | 18.90 | 48.03 | 16.44 |
| **NeuroKalman** | **71.56** | **25.86** | **58.73** | **22.43** | **42.70** | **30.52** | **62.70** | **25.86** | **105.07** | **20.11** | **53.90** | **18.21** |
| *Test UnSeen Map Set* | | | | | | | | | | | | |
| MBGRU | 104.04 | 7.52 | 31.21 | 6.42 | 75.90 | 8.20 | 36.36 | 6.87 | 143.83 | 6.55 | 23.93 | 5.79 |
| **NeuroKalman** | **100.32** | **8.34** | **34.15** | **7.12** | **69.50** | **9.15** | **38.50** | **7.50** | **140.00** | **7.20** | **28.00** | **6.50** |
| *Test UnSeen Object Set* | | | | | | | | | | | | |
| MBGRU | 75.16 | 31.00 | 56.28 | 26.68 | 50.40 | 41.15 | 63.64 | 34.88 | 87.49 | 25.95 | 52.62 | 22.60 |
| **NeuroKalman** | **71.01** | **32.48** | **60.82** | **28.50** | **44.50** | **42.50** | **66.50** | **37.37** | **84.50** | **27.50** | **58.00** | **24.50** |

*Table 9.* Auxiliary loss supervision. All methods are implemented with the L1 assistant on the UAV-Need-Help test seen set.

| Methods | Full | | | | Easy | | | | Hard | | | |
|---|---|---|---|---|---|---|---|---|---|---|---|---|
| | NE↓ | SR↑ | OSR↑ | SPL↑ | NE↓ | SR↑ | OSR↑ | SPL↑ | NE↓ | SR↑ | OSR↑ | SPL↑ |
| w/o Aux Loss | 80.56 | 25.39 | 53.17 | 21.67 | 51.70 | 30.52 | 56.70 | 25.86 | 119.17 | 19.06 | 48.82 | 17.17 |
| **w/ Aux Loss** | **71.56** | **25.86** | **58.73** | **22.43** | **42.70** | 30.52 | **62.70** | 25.86 | **105.07** | **20.11** | **53.90** | **18.21** |

## A.2. Additional Results

Our core hypothesis is that the Bayesian structural decoupling is prerequisites for solving state drift. For this, we provide additional experiments about architecture topology. Furthermore, we validate that the augmented supervision is beneficial for waypoint prediction.

**Effect of Structural Decoupling Architecture.** We validate the structural advantage of our Bayesian formulation by comparing it with a coupled temporal modeling baseline (MBGRU), where memory-enhanced features are directly fed into the GRU for waypoint prediction. As shown in Table 8, this baseline suffers a substantial performance drop on the Hard split. This indicates that without explicit separation, the network treats memory only as context, not as a corrective anchor. As a result, the GRU over-relies on its motion dynamic prediction, leading to severe drift over long horizons.

**Auxiliary Loss Supervision.** We also analyze the impact of applying additional loss supervision to the intermediate Prior ($\tilde{\mathbf{z}}_t$) and Measurement Likelihood ($\mathbf{r}_t$). As shown in Table 9, we observe that removing these auxiliary regression losses leads to a notable increase in NE (80.56 vs. 71.56) and OSR (53.17% vs. 58.73%). This suggests that explicit supervision is crucial to prevent module collapse; it forces the GRU to strictly learn transition dynamics and the measurement to learn grounding, thereby ensuring the interpretability and robustness of the Kalman correction against structural degradation.

## A.3. Qualitative Trajectory Visualization

We further analyze a specific long-range navigation scenario in Figure 5 (Top-down view) and Figure 6 (Front view) to diagnose the failure modes of the baseline. From the Top-Down perspective (Fig. 5), the TravelUAV-FT suffers from severe disorientation. Due to accumulated state drift, it fails to register the spatial relationship between its position and the key landmarks, ultimately losing its bearing and failing to locate the target.

The Front-View sequence (Fig. 6) offers a complementary explanation for this failure. While TravelUAV-FT initially adheres to the planned route, it exhibits rigid maneuverability. Even when visual landmarks are visible, the model lacks the dynamic flexibility to adjust its flight trajectory in time. Consequently, it overshoots the critical turning point, progressively drifts away from the target as time elapses, and eventually collides with obstacles. In contrast, NeuroKalman successfully anchors its state estimate to visual landmarks. By dynamically correcting its belief state, our model makes a clear turn at the decision point, which stops small errors from accumulating over time and allows the UAV to reach the target safely.

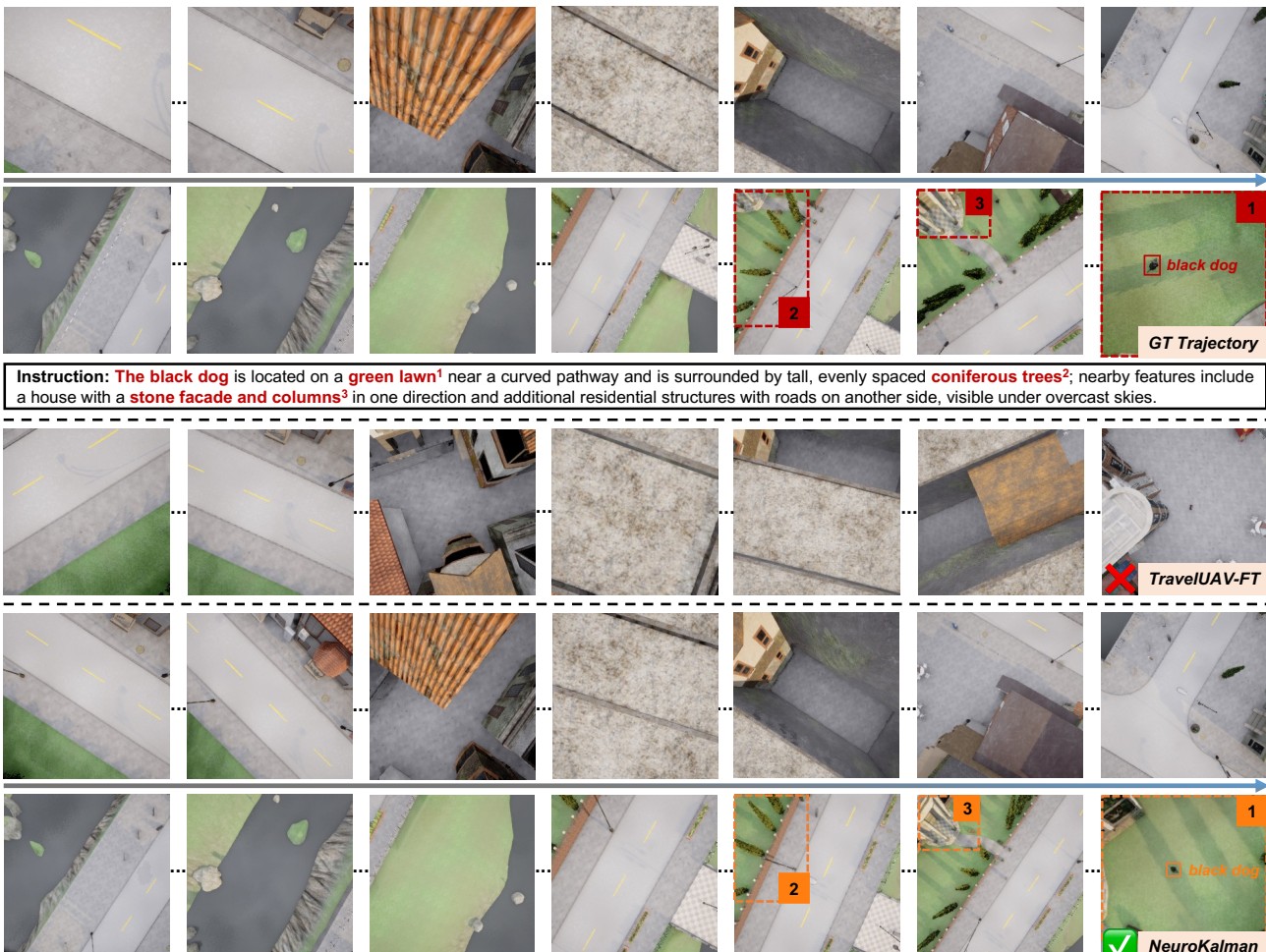

**Instruction:** **The black dog** is located on a **green lawn**[1] near a curved pathway and is surrounded by tall, evenly spaced **coniferous trees**[2]; nearby features include a house with a **stone facade and columns**[3] in one direction and additional residential structures with roads on another side, visible under overcast skies.

*Figure 5.* **Navigation example comparison between the TravelUAV-FT and our NeuroKalman (Top-Down View).** Due to severe state drift, TravelUAV-FT fails to recognize key landmarks and loses its orientation, resulting in a failed search. In contrast, NeuroKalman successfully anchors its position against structural features, maintaining the correct heading towards the target.

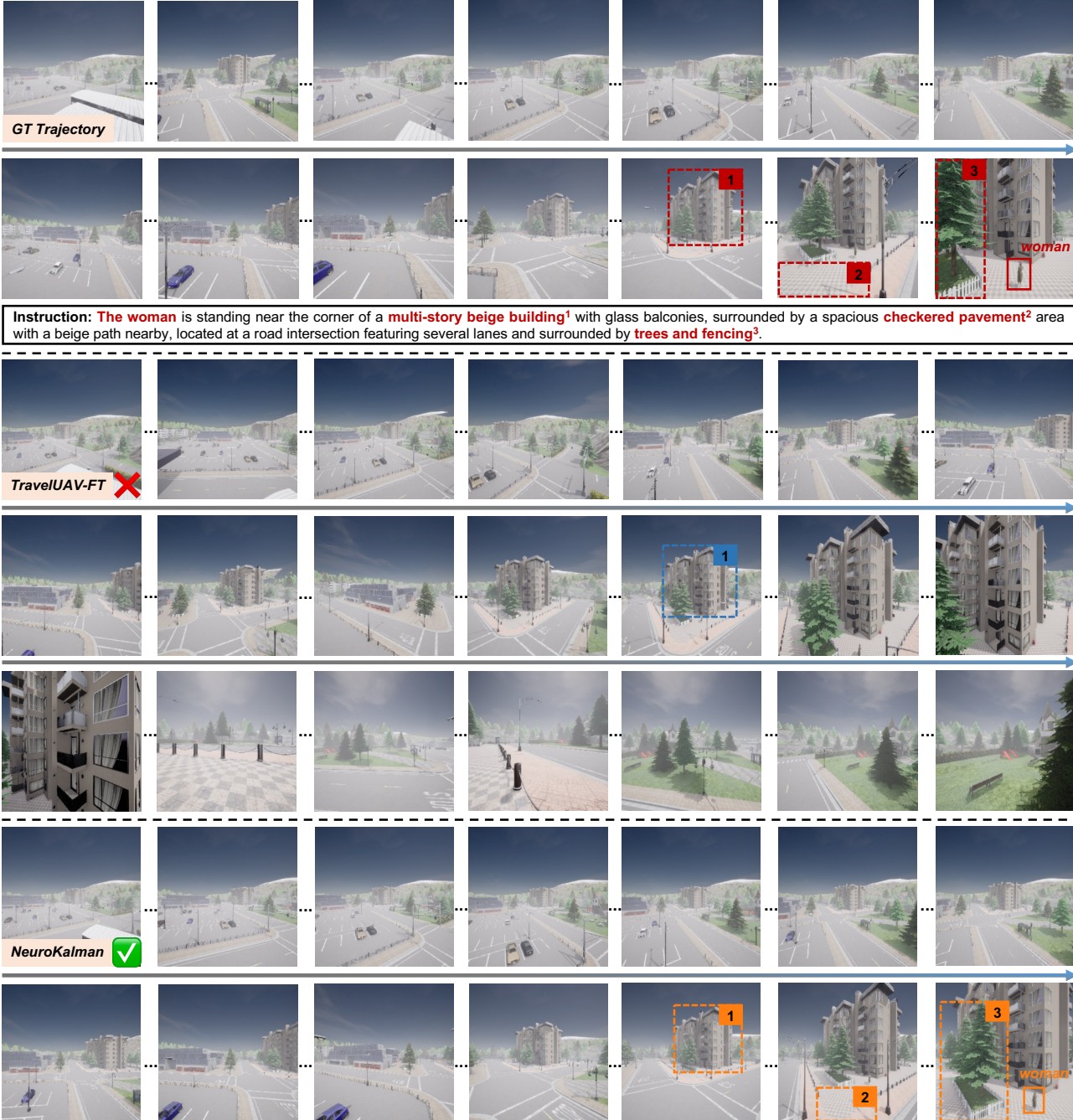

**Instruction:** The woman is standing near the corner of a **multi-story beige building**[1] with glass balconies, surrounded by a spacious **checkered pavement**[2] area with a beige path nearby, located at a road intersection featuring several lanes and surrounded by **trees and fencing**[3].

*Figure 6.* **Navigation example comparison between the TravelUAV-FT and our NeuroKalman (Front View).** TravelUAV-FT lacks the maneuverability to adjust its trajectory upon detecting landmarks, eventually missing the target and drifting into a collision. Conversely, NeuroKalman leverages memory-augmented updates to execute precise turning maneuvers.

