# OpenReview forum: "Mitigating Error Accumulation in Continuous Navigation via Memory-Augmented Kalman Filtering"
_ICML.cc/2026/Conference — ICML 2026 regular_

### Official Review · Reviewer_PUeS · 2026-02-19

**Soundness:** 3
**Presentation:** 3
**Significance:** 3
**Originality:** 3
**Overall Recommendation:** 4
**Confidence:** 4

**Summary:**

This paper addresses the problem of error accumulation in continuous UAV vision-language navigation (VLN). Existing models rely on recurrent dead-reckoning style prediction, where the next waypoint is iteratively predicted based on previous estimates. This leads to accumulated positional error and state drift over long horizons.

To mitigate this, the authors propose NeuroKalman, a memory-augmented Kalman-style correction framework. The core idea is to reframe navigation as recursive Bayesian state estimation:
- **Prediction step**: A GRU-based recurrent model predicts a prior latent belief state.
- **Update step**: A memory bank stores reliable historical visual anchors. Retrieval over this memory is interpreted as kernel density estimation and implemented via attention.
- **Kalman correction**: A learnable gating mechanism estimates a Kalman gain to fuse prior and measurement in latent space.

The method is evaluated on the TravelUAV benchmark. Notably, the model is pretrained on full data and then fine-tuned using only 10% of the training set. The authors claim improved data efficiency and stronger long-horizon robustness, especially in hard splits (>250m trajectories). Error curves (Figure 4) suggest that the proposed method stabilizes or reduces accumulated L2 error after long sequences, whereas baselines show monotonic drift.

**Compliance With Llm Reviewing Policy:**

Affirmed.

**Key Questions For Authors:**

- **Data Efficiency Claim Clarification.**
The model is pretrained on the full dataset and then fine-tuned on 10% of the training data. Could the authors clarify whether NeuroKalman trained from scratch on 10% of data still outperforms baselines?
If performance remains competitive under true low-data training, this would significantly strengthen the data efficiency claim. Otherwise, the claim may need reframing as fine-tuning robustness rather than data efficiency.
- **Short- vs Long-Horizon Trade-off.**
The correction mechanism appears to become beneficial primarily after substantial drift accumulation (e.g., beyond ~ 80 steps). Have the authors conducted a separate short-horizon analysis to determine whether the correction introduces noise or over-regularization in early stages?
A clearer characterization of this trade-off would help assess the method’s robustness across trajectory lengths.

**Limitations:**

The authors partially discuss limitations, particularly regarding the use of a GRU-based prior and the need for safety verification before real-world deployment. However, several important limitations are not fully addressed.

First, the claim of data efficiency could be more carefully framed, given that the model is pretrained on the full dataset before fine-tuning on 10%. Second, the potential trade-off between short- and long-horizon performance is not explicitly analyzed.

While the paper includes an impact statement acknowledging safety considerations, a deeper discussion of these methodological limitations would further strengthen the work.

**Strengths And Weaknesses:**

**Strengths**
- The core motivation—error accumulation in recurrent UAV navigation—is well-founded and clearly articulated.
- The decomposition into prediction (prior) and correction (likelihood) is conceptually consistent with Bayesian filtering.
- Empirical results show consistent improvements on long trajectories and hard splits.

**Weaknesses**
- The claimed “data efficiency” is potentially overstated. The model is pretrained on full data and only fine-tuned on 10%. This demonstrates robustness to limited fine-tuning rather than genuine low-data learning.
- Short-horizon performance is not separately analyzed. The correction mechanism appears beneficial only after significant drift accumulates (e.g., after ~100 steps), suggesting a possible trade-off or noise injection in early stages.

---

> ### Author Rebuttal · Authors · 2026-03-31
>
> > **W1: "The claimed “data efficiency”, ..., genuine low-data learning."** **Q1: "Data Efficiency Claim Clarification, ..., fine-tuning robustness rather than data efficiency."**
>
> **A1:** Our model demonstrates its data efficiency by effectively modeling temporal context under two training settings: (i) training from scratch using only 10% of the data, and (ii) fine-tuning from pre-trained weights with 10% data.
>
> | Setting (L1 Test-Seen) | NE↓ | SR↑ | OSR↑ | SPL↑ |
> |:---|:---:|:---:|:---:|:---:|
> | TravelUAV (100% training data) | 106.28 | 16.10 | **44.26** | 14.30 |
> | TravelUAV (no full-data pretrain, 10%) | 122.63 | 13.19 | 37.16 | 11.71 |
> | NeuroKalman (no full-data pretrain, 10%) | **86.37** | **22.00** | 43.79 | **18.46** |
>
> As shown, NeuroKalman trained with only 10% data not only significantly outperforms TravelUAV under the identical 10% setting (29.6% lower NE, 66.8% higher SR), but also remarkably surpasses the fully trained TravelUAV (100% data) across primary metrics. Meanwhile, TravelUAV only predicts waypoints step-by-step from current observations without explicit long-horizon error correction, leaving its performance heavily bottlenecked by the MLLM’s inherent capacity and resulting in similarly low performance under both 10% and 100% data. By contrast, **NeuroKalman breaks this bottleneck via explicit error correction and temporal modeling, demonstrating its low-data efficiency.**
>
> > **W2: "The Short-horizon performance, ..., noise injection in early stages."** **Q2: "Short- vs Long-Horizon Trade-off, ..., across trajectory lengths."**
>
> **A2:** Our correction mechanism proves highly robust for both short- and long-horizon navigation across seen and unseen environments. Specifically, as shown in Figure 4 in the original paper, while differing training setups cause a slight performance gap compared to the fully-trained TravelUAV in short horizons, our approach comprehensively surpasses TravelUAV-FT in both short- and long-horizon navigation under an identical 10% fine-tuning. More importantly, we performed the truncated-horizon evaluation using the Average Displacement Error over the first K steps (ADE@K, mean±std) and demostrate that our method yields the best performance across short- and long-horizon tasks in unseen environments.
>
> | Method | ADE@20 | ADE@40 | ADE@60 | ADE@80 | ADE@100 |
> |:---|:---:|:---:|:---:|:---:|:---:|
> | TravelUAV | 7.54±8.18 | 10.72±5.76 | 15.18±9.28 | 18.47±9.73 | 14.08±5.14 |
> | TravelUAV-FT | 9.84±5.16 | 13.32±8.55 | 17.16±11.40 | 21.40±10.10 | 25.55±7.76 |
> | NeuroKalman | **6.61±4.80** | **8.70±6.22** | **11.41±7.75** | **12.12±5.36** | **8.02±4.39** |
>
> As shown, on the unseen set, NeuroKalman dominates both baselines across all horizons, maintaining tightly bounded errors. We also provide a visualization of L2 position error over time on the unseen set in this [link](https://anonymous.4open.science/r/Neuro-Kalman-Rebuttal-17D3/L2_unseen.png). This indicates that the core advantage of **NeuroKalman is not merely improved short-term prediction accuracy, but rather its ability to more stably suppress cumulative drift over longer trajectories**, especially under distribution shift conditions, which is consistent with the superior results on the unseen benchmark for UO/UM in the original paper.
>
> > **L1: "The authors partially discuss limitations, ..., would further strengthen the work."**
>
> **A3:** We have addressed this in the corresponding responses. Specifically, please refer to **_W1/Q1-A1_** for the claim of data efficiency, **_W2/Q2-A2_** for the potential trade-off between short- and long-horizon performance.

---

> > ### Author Rebuttal · Reviewer_PUeS · 2026-04-01
> >
> > Thank you for the detailed rebuttal and for providing additional experimental results. The new evidence is helpful and partially addresses some of the concerns raised in the original review. In particular, the added low-data and truncated-horizon analyses provide useful clarification of the method’s behavior.
> >
> > That said, I do not think all of the limitations are fully resolved yet. The low-data efficiency claim would be more convincing with clearer details on how the 10% subset was constructed and whether the results are stable across different samplings or random seeds. In addition, while the ADE@K results are helpful, they do not directly address the concern about possible instability in the early stages of long-horizon trajectories. Overall, the rebuttal improves the paper, but some of the original limitations remain only partially addressed.
> >
> > ## Additional Questions
> >
> > 1. **Fairness of the 10% training setting**
> >    Thank you for the additional low-data results. However, the data-efficiency claim would be easier to assess if the experimental protocol were described more clearly. In particular, it is still unclear how the 10% subset was constructed. Was it sampled randomly, stratified by trajectory length or difficulty, or selected using some other criterion? Since low-data performance can vary substantially depending on the subset composition and random seed, please clarify the sampling procedure and whether the reported results are stable across different subset selections.
> >
> > 2. **Cause of the early-stage instability in long-horizon trajectories**
> >    The added ADE@K results are helpful, but they do not fully address the concern about the behavior in the early phase of long-horizon trajectories. In the original error curves, the method appears somewhat unstable in the earlier portion of long trajectories before the correction mechanism becomes clearly beneficial. Could the authors elaborate on the possible cause of this behavior? For example, is it related to noisy memory retrieval, delayed correction dynamics, or a mismatch between the prior prediction and the update step in the early stage? A clearer explanation of this phenomenon would help better characterize both the strengths and the limitations of the proposed method.

---

> > > ### Author Response · Authors · 2026-04-03
> > >
> > > We thank the reviewer for the follow-up comments and are happy to provide further clarification from the following two aspects.
> > >
> > > > **Q1: "Fairness of the 10% training setting."**
> > >
> > > **A1:** (1) _Subset Construction Protocol._  We perform uniform random sampling at the trajectory level using fixed random seeds. This strategy preserves the original data distribution without introducing subset bias. Specifically, our randomly sampled subset (793 trajectories across 19 scenes) consists of 421 "Easy" (<=250m) and 372 "Hard" (>250m) trajectories. This 53.1% to 46.9% ratio aligns with the difficulty distribution of the full dataset (52.4% to 47.6%), **indicating that the 10% subset provides a representative snapshot of the overall data.**
> > >
> > > | Model Setting (L1 Test-Seen) | Easy/Hard Ratio | NE↓ | SR↑ | OSR↑ | SPL↑ |
> > > | :--- | :---: | :---: | :---: | :---: | :---: |
> > > | TravelUAV (100% Full Data) | 52.4%/47.6% | 106.28 | 16.10 | 44.26 | 14.30 |
> > > | NeuroKalman (10% Data, Seed 1) | 53.1%/46.9% | 86.37 | 22.00 | 43.79 | 18.46 |
> > > | NeuroKalman (10% Data, Seed 2) | 51.7%/48.3% | 88.33 | 23.67 | 45.64 | 19.82 |
> > > | NeuroKalman (10% Data, Seed 3) | 50.7%/49.3% | 82.97 | 22.42 | 45.70 | 18.85 |
> > > | **NeuroKalman (10% Avg ± Std)** | **51.8%/48.2%** | **85.89±2.71** | **22.70±0.87** | **45.04±1.09** | **19.04±0.70** |
> > >
> > > (2) _Stability Across Different Random Seeds._ The reported results of NeuroKalman are stable across subset selections of different random seeds. As shown in the table above, we generated multiple 10% subsets using different random seeds (e.g., Seeds 1, 2, and 3). The results show that NeuroKalman exhibits marginal performance variance across different random splits and consistently outperforms the fully trained TravelUAV. **This indicates that our model's low-data efficiency does not stem from subset composition, but is driven by the model's explicit long-horizon error correction and robust temporal modeling capabilities.**
> > >
> > > > **Q2: "Cause of the early-stage instability in long-horizon trajectories."**
> > >
> > > **A2:** The early-stage instability is inherently caused by the simple prediction of the GRU—a preliminary baseline architecture for the prediction block. Specifically, NeuroKalman fuses the prior and observation via a standard Bayesian update:
> > >
> > > $\mathbf{z}_t = (1 - \mathbf{K}_t) \odot \tilde{\mathbf{z}}_t + \mathbf{K}_t \odot \mathbf{r}_t$,
> > >
> > > where $\mathbf{z}_t$ is the final updated vector, $\mathbf{K}_t$ is the adaptive Kalman gain, $\tilde{\mathbf{z}}_t$ is the vector predicted by GRU, and $\mathbf{r}_t$ is the memory-augmented observation vector. The $\mathbf{K}_t$ learned in our mechanism starts small due to limited visual context, i.e., $\tilde{\mathbf{z}}_t$ is highly weighted in the final vector that is directly used for waypoint prediction. Since fine-tuning from limited 10% data inevitably introduces prediction instability, both TravelUAV-FT and our method exhibit position error fluctuations during this prior-dependent period. However, as navigation progresses, the memory module gradually retrieves more visual anchors, leading to an increase in $\mathbf{K}_t$. This shifts the model **from prior-dependent prediction to observation-aware correction, allowing the reliable $\mathbf{r}_t$ to progressively suppress accumulated errors.**
> > >
> > > We further support this with empirical evidence from [Figure (a)-(b)](https://anonymous.4open.science/r/Neuro-Kalman-Rebuttal-17D3/trajectory_analysis.png). Figure (a) shows the 3D navigation trajectories comparison, where NeuroKalman guides the trajectory back onto the correct manifold via Kalman correction, thereby mitigating the deviation observed in TravelUAV baseline. Figure (b) illustrates the temporal dynamics of $\mathbf{K}_t$, which exhibits a clear increasing trend over time, confirming the transition from prior-dependent prediction in the early stages toward observation-guided correction in the later stages. We note the inherent vulnerability of the simple GRU prediction under the low-data setting and will investigate the advanced architectures to effectively mitigate the early-stage instability.

---

### Official Review · Reviewer_zXza · 2026-03-12

**Soundness:** 2
**Presentation:** 3
**Significance:** 2
**Originality:** 3
**Overall Recommendation:** 3
**Confidence:** 3

**Summary:**

This paper targets state drift / error accumulation in continuous UAV vision-language navigation (VLN), where step-by-step dead-reckoning causes the agent’s internal belief to misalign with true coordinates over long horizons. They proposed NeuroKalman which reframes navigation as recursive Bayesian filtering over a latent belief state: a Prediction step (GRU prior / motion dynamics) is combined with an Update step (measurement likelihood from an MLLM that uses retrieved historical observations). The authors interpret attention-based retrieval as kernel density estimation (KDE) over visual features and use a learnable Kalman gain to fuse prior and measurement in latent space.

**Compliance With Llm Reviewing Policy:**

Affirmed.

**Key Questions For Authors:**

- What happens if the model is trained without full-data pretraining (e.g., from scratch or from a generic pretrained model) using only 10% of TravelUAV? Does NeuroKalman still show a clear advantage?
- What is the inference-time cost (latency + memory footprint) of the MLLM Update + retrieval + gain computation per step? Is this compatible with real-time UAV control loops?
- How sensitive is performance to the confidence threshold for memory insertion (e.g., 𝜎_𝑡>0.5) and to retrieval noise? Are there failure cases where memory hurts?

**Limitations:**

Yes, The paper briefly acknowledges that its GRU-based prediction prior may suffer from information decay over very long horizons, but it does not discuss other important limitations such as computational overhead of the MLLM and memory retrieval, scalability of the memory bank, sensitivity to confidence thresholds, potential retrieval errors,..etc

**Strengths And Weaknesses:**

### Strengths
- A strong motivation such that long-horizon drift is a real failure mode for continuous VLN; the paper directly attacks it rather than only improving the next-step predictors.
- Explicitly separating a motion prior from a correction step is a clean conceptual framing, and the Kalman-style fusion makes the approach more interpretable than “just add memory.”
- Experiments show consistent gains on seen/unseen settings; there are ablations comparing fixed vs learnable gains and memory length, plus drift-over-time visualization.
### Weakness
- The Update step uses a frozen MLLM backbone (EVA-CLIP + Vicuna-7B) with LoRA and retrieval. The paper does not clearly report inference latency, memory cost, or whether this is feasible for onboard UAV compute.
- The paper states that both NeuroKalman and the TravelUAV baseline are initialized from weights pre-trained on the full data and then fine-tuned on 10%. That is not the same as learning with only 10% data from scratch, and it weakens the claim that the approach fundamentally needs less data.
- All results are on TravelUAV (AirSim). There are no real-world UAV tests, and generalization claims are limited to that benchmark’s splits.
The authors note the GRU prior may suffer information decay on very long horizons, but other limitations (memory scaling, retrieval brittleness, compute constraints) aren’t discussed in comparable depth.
- The method depends on (i) retrieval relevance and (ii) confidence calibration (𝜎_𝑡) that controls what gets stored. If confidence is miscalibrated, the memory bank could accumulate false anchors and reinforce drift. The paper does not deeply analyze worst-case retrieval errors or safeguards.

---

> ### Author Rebuttal · Authors · 2026-03-31
>
> > **W1: "The Update step, ..., onboard UAV compute."**  **Q2: "What is the inference-time cost, ..., UAV control loops?"**
>
> **A1:** Our method is feasible for real-world UAV deployment under the standard ground–drone collaborative paradigm. We further profiled the inference-time cost of NeuroKalman and TravelUAV baseline on a single NVIDIA A6000 GPU to highlight this.
>
> | Method | Latency / step (s) | GPU Memory (GB) |
> | :--- | :---: | :---: |
> | **TravelUAV** | 0.409 | 19.5 |
> | **NeuroKalman** | 0.488 | 23.3 |
> | *MLLM Update* | 0.455 | 19.5 |
> | *Memory Retrieval* | 0.021 | 2.6 |
> | *Kalman Gain Computation* | 0.012 | 1.2 |
>
> **The incremental overhead is minimal.** The model predicts waypoints for a 5-meter flight within 0.5s, shorter than physical execution time at 2 m/s. Following common practice, UAV MLLM systems adopt a ground-drone collaborative paradigm (e.g., UAV-Flow [1]), offloading heavy inference to a ground station. Our method imposes no extra heavy overhead, **enabling real-time navigation on server-grade platforms.**
>
> [1] Wang Xiangyu, et al. "Uav-flow colosseo: A real-world benchmark for flying-on-a-word uav imitation learning."
>
> > **W2: "The paper states that, ..., needs less data."** **Q1: "What happens if the model, ..., still show a clear advantage?"**
>
> **A2:** Our model demonstrates data efficiency under two settings: (i) training from scratch on 10% data, and (ii) fine-tuning from pre-trained weights with 10% data.
>
> | Setting (L1 Test-Seen) | NE↓ | SR↑ | OSR↑ | SPL↑ |
> |:---|:---:|:---:|:---:|:---:|
> | TravelUAV (100% training data) | 106.28 | 16.10 | **44.26** | 14.30 |
> | TravelUAV (no full-data pretrain, 10%) | 122.63 | 13.19 | 37.16 | 11.71 |
> | NeuroKalman (no full-data pretrain, 10%) | **86.37** | **22.00** | 43.79 | **18.46** |
>
> NeuroKalman with only 10% data significantly outperforms TravelUAV under the identical 10% setting (29.6% lower NE, 66.8% higher SR) and surpasses the fully trained TravelUAV (100% data) across primary metrics. TravelUAV predicts waypoints step-by-step without explicit error correction, leaving performance bottlenecked by the MLLM's capacity. **NeuroKalman breaks this bottleneck via explicit error correction and temporal modeling, demonstrating its low-data efficiency.**
>
> > **W3: "All results are on TravelUAV, ..., aren't discussed in comparable depth."**
>
> **A3:** (1) _Real-world UAV Test._ We fully agree with the importance of real-world UAV validation and would love to conduct physical flight experiments. However, achieving reliable long-horizon navigation in high-fidelity simulation is a prerequisite for safe deployment. TravelUAV provides diverse realistic settings where our method achieves strong performance. Recent methods like CityNav [1] and APEX [2] still exhibit poor success rates (8.69/12.36) even under simplified conditions, highlighting fundamental algorithmic gaps that must be bridged first.
>
> [1] Jungdae Lee, et al. "CityNav: Language-Goal Aerial Navigation Dataset with Geographic Information."
>
> [2] Zhang Daoxuan, et al. "APEX: A Decoupled Memory-based Explorer for Asynchronous Aerial Object Goal Navigation."
>
> (2) _Others._ **Retrieval brittleness:** our ablation shows over-relying on measurement (K_t=0.9) degrades performance, which is why we introduced adaptive Kalman-gain (Page 8, Line 425). **Memory scaling:** extending memory from M=10 to M=15 hurts performance due to outdated features acting as noise (Page 8, Line 398). **Compute constraints:** addressed in **W1-A1** above.
>
> > **W4: "The method depends on (i) retrieval relevance and, ..., retrieval errors or safeguards."** **Q3: "How sensitive is performance, ..., failure cases where memory hurts?**
>
> **A4:** Our approach robustly handles retrieval noise and there are no failure cases where the memory mechanism hurts performance. While overly low thresholds (e.g., σ_t=0.3) cause minor noise, our safeguard can be directly derived (σ_t >= 0.5) and reliably blocks false anchors, preventing the memory-induced drift.
>
> | σ_t | NE↓ | SR↑ | OSR↑ | SPL↑ |
> |:---|:---:|:---:|:---:|:---:|
> | 0.3 | 82.45 | 20.50 | 48.15 | 17.22 |
> | 0.5 | **71.56** | **25.86** | **58.73** | **22.43** |
> | 0.7 | 73.88 | 24.55 | 55.62 | 22.10 |
> | 0.9 | 75.45 | 24.18 | 53.30 | 21.75 |
>
> NeuroKalman remains stable for σ_t ≥ 0.5, with σ_t = 0.5 yielding the best performance. A low threshold (σ_t = 0.3) hurts by admitting noisy anchors. **Our method exhibits a robust threshold range:** the memory is inserted only through post-correction selective storage, and the posterior is filtered via confidence-aware Kalman fusion, ensuring noisy retrievals are not unconditionally trusted.
>
> > **L1: "Yes, The paper briefly acknowledges that its GRU-based prediction prior, ..., potential retrieval errors, etc**
>
> **A5:** We have addressed this in the corresponding responses. Specifically, please refer to **_W1/Q2-A1_** for computational overhead, **_W4/Q3-A4_** for threshold sensitivity and retrieval errors, and **_W3-A3_** for memory scalability.

---

### Official Review · Reviewer_q4ky · 2026-03-29

**Soundness:** 2
**Presentation:** 3
**Significance:** 2
**Originality:** 2
**Overall Recommendation:** 3
**Confidence:** 3

**Summary:**

The paper proposes NeuroKalman, a framework that applies recursive Bayesian estimation (Prediction + Update) to mitigate error accumulation in UAV VLN. The authors claim that combining a GRU-based prior with an attention-based memory bank via a learnable Kalman gain improves robustness, especially under low-data paradigms.

**Compliance With Llm Reviewing Policy:**

Affirmed.

**Key Questions For Authors:**

Compared to other traditional Kalman models or Kalman networks, what are the advantages of the proposed paradigm? Comparison based on inference latency, memory usage, and accuracy.

**Limitations:**

Without demonstrations on real-world scenarios and devices, the effectiveness and generalization ability of the method cannot be truly verified.

**Strengths And Weaknesses:**

Strengths:

- Using recurrent Bayesian filtering of latent belief states, combining prediction and update steps, to correct long-range errors presents a clear and effective framework.
- Experiments on the TravelUAV benchmark show that, fine-tuned using only 10% of the data, this method significantly outperforms existing baselines in unseen map and object segmentation.

Weaknesses:
- The method uses an MLLM model and related steps for inference, but the paper does not report inference latency, memory usage, or how inference is performed on real drones. I understand that this paradigm may not be usable on real-world platforms.
- All experimental results are only validated on the simulated TravelUAV dataset; no validation or demonstration on real machines is provided, making it impossible to determine the method's generalization ability in real-world scenarios.
- About data efficiency, this method is pre-trained on the full dataset and then fine-tuned on the 10% dataset. Is there a problem with this data efficiency description?

---

> ### Author Rebuttal · Authors · 2026-03-31
>
> > **W1: "The method uses an MLLM model and, ..., be usable on real-world platforms."**
>
> **A1:** Our method is feasible for real-world UAV deployment under the standard ground–drone collaborative paradigm [1]. We further profiled the inference-time cost of NeuroKalman and TravelUAV baseline on a single NVIDIA A6000 GPU to highlight this.
>
> | Method | Latency / step (s) | GPU Memory (GB) |
> |:---|:---:|:---:|
> | TravelUAV | 0.409 | 19.5 |
> | NeuroKalman | 0.488 | 23.3 |
>
> (1) _Inference Cost._ As shown, **the incremental overhead introduced by NeuroKalman is minimal.** Using a single NVIDIA A6000 GPU with bf16 precision, the average forward-pass time is 0.488 seconds per step. Given that the model predicts trajectory waypoints for a 5-meter flight within 0.5 seconds, the inference time is significantly shorter than the physical execution time (e.g., at a safe flight speed of 2 m/s).
>
> (2) _Real Drones Deployment._ Current UAV MLLM systems mostly adopt a ground-drone collaborative paradigm (e.g., UAV-Flow [1]) for real-world inference. The UAV utilizes a lightweight edge module for flight control and data streaming, while heavy inference is offloaded to a ground station via a low-latency wireless link. As our method imposes no extra heavy computational overhead relative to existing works, **it enables real-time navigation on server-grade platforms.**
>
> [1] Wang Xiangyu, et al. "Uav-flow colosseo: A real-world benchmark for flying-on-a-word uav imitation learning."
>
> > **W2: "All experimental results, ..., generalization ability in real-world scenarios."**
>
> **A2:** We fully agree with the importance of real-world validation. However, achieving reliable long-horizon navigation in high-fidelity simulation is a prerequisite for safe deployment. TravelUAV provides diverse realistic settings where our method achieves strong performance. Recent methods like CityNav [1] and APEX [2] still exhibit poor success rates (8.69/12.36) under simplified conditions, highlighting fundamental gaps to bridge before real-world deployment.
>
> [1] Jungdae Lee, et al. "CityNav: Language-Goal Aerial Navigation Dataset with Geographic Information."
>
> [2] Zhang Daoxuan, et al. "APEX: A Decoupled Memory-based Explorer for Asynchronous Aerial Object Goal Navigation."
>
> > **W3: "About data efficiency, ..., data efficiency description?"**
>
> **A3:** Our model demonstrates data efficiency under two settings: (i) training from scratch on 10% data, and (ii) fine-tuning from pre-trained weights with 10% data.
>
> | Setting (L1 Test-Seen) | NE↓ | SR↑ | OSR↑ | SPL↑ |
> |:---|:---:|:---:|:---:|:---:|
> | TravelUAV (100% training data) | 106.28 | 16.10 | **44.26** | 14.30 |
> | TravelUAV (no full-data pretrain, 10%) | 122.63 | 13.19 | 37.16 | 11.71 |
> | NeuroKalman (no full-data pretrain, 10%) | **86.37** | **22.00** | 43.79 | **18.46** |
>
> As shown, NeuroKalman with only 10% data outperforms TravelUAV under the identical 10% setting (29.6% lower NE, 66.8% higher SR) and surpasses the fully trained TravelUAV (100% data). TravelUAV predicts waypoints step-by-step without explicit error correction, bottlenecked by MLLM capacity. **NeuroKalman breaks this bottleneck via explicit error correction and temporal modeling, demonstrating its low-data efficiency.**
>
> > **Q1: "Compared to other traditional Kalman models, ..., memory usage, and accuracy."**
>
> **A4:** Our proposed NeuroKalman performs correction in a multimodal latent space, enabling strong cross-modal fusion and generalization, thereby overcoming the limitation of traditional Kalman methods that only operates over discrete waypoints. We implemented a baseline (TravelUAV + Post-KF) applying a standard constant-velocity Kalman filter [1] to the 4D waypoint output (8D state: 4D waypoint + 4D velocity). The results on L1 Test-Seen are as follows:
>
> | Method | NE↓ | SR↑ | OSR↑ | SPL↑ | Latency | Memory |
> |:---|:---:|:---:|:---:|:---:|:---:|:---:|
> | TravelUAV | 106.28 | 16.10 | 44.26 | 14.30 | 0.409 s | 19.5 GB |
> | TravelUAV + Post-KF | 96.67 | 18.17 | 33.65 | 15.64 | 0.411 s | 19.5 GB |
> | NeuroKalman | **71.56** | **25.86** | **58.73** | **22.43** | 0.488 s | 23.3 GB |
>
> As shown, the TravelUAV (with Post-KF) yields only marginal improvements as it smooths geometric noise without access to history context. **NeuroKalman delivers substantial gains (SR 16.10→25.86) via latent-space correction where the GRU prior and attention-based memory enable semantically informed error correction.** While NeuroKalman adds +77 ms latency and +3.8 GB memory, it achieves a far better accuracy-efficiency trade-off.
>
> [1] Guy Revach et al. "KalmanNet: Neural network aided Kalman filtering for partially known dynamics."
>
> > **L1: "Without demonstrations on real-world scenarios, ..., method cannot be truly verified."**
>
> **A5:** We have addressed this in the corresponding responses. Specifically, please refer to **_W2-A2_**.

---

### Decision · Program_Chairs · 2026-04-30

**Decision:**

Accept (regular)

**Comment:**

The paper initially received mixed reviews, one weak accept and one weak reject.  ( The third reviewer was unable to provide a review in time).  After rebuttal, both reviewers confirmed that their concerns have been (though partially) addressed.  R#1 further asked some following-up questions to which the authors provided detailed responses. Although no further round of discussions were conducted, the remaining concerns seem to be addressable and the paper rescuable.   With all information gathered, the  SAC considers the paper is acceptable.